# Rethinking Druggability in the Evaluation of AI-driven Structure-based Drug Design

## Abstract

Structure-based drug design harnesses three-dimensional structural information to guide ligand discovery and has seen rapid progress through machine learning. Yet the evaluation of AI-driven SBDD models has largely ignored **druggability**—the propensity of a binding pocket to accept a small, drug-like molecule. As a result, generative models may appear successful by creating compounds that dock well to pockets that are not feasible drug targets. We review SBDD benchmarks and druggability assessment methods, highlight pitfalls of current evaluation protocols, and propose a methodology to incorporate continuous druggability scores into the widely used CrossDocked2020 benchmark. By weighting generative scores according to pocket druggability and analysing performance across druggable and undruggable targets, our framework encourages models to focus on realistic therapeutic targets and reveals algorithmic biases.

## 1   Introduction

**Structure-based drug design (SBDD)** has become a cornerstone of modern drug discovery because it directly leverages the three-dimensional (3D) structure of a target to find ligands with complementary shape, electrostatics, and hydrophobicity. Compared with high-throughput screening, SBDD provides a more targeted and cost-efficient approach for lead generation. Reviews of the field note that SBDD is "becoming an essential tool for faster and more cost-efficient lead discovery" and that it is widely used in industry and academia [4]. The availability of high-resolution structures for thousands of proteins and the rapid advances in machine learning make it possible to automate key steps such as virtual screening, docking and ligand optimization. In recent years, **AI-driven *de novo*** design models have emerged that attempt to generate novel small molecules tailored to a given pocket, often relying on geometric deep learning to encode the pocket's shape and chemical environment. These models promise to accelerate drug discovery but require careful evaluation.

**Druggability** refers to the propensity of a protein binding site to bind drug-like small molecules with high affinity. A binding pocket may be druggable because of its size, depth and hydrophobicity; a "druggable pocket" is one where small drug-like molecules have been shown to bind [28]. Distinguishing *druggability* from related concepts is important: *ligandability* measures whether a site can bind any small molecule, whereas druggability implies the ability to modulate a target to achieve a therapeutic effect [12]. Only about a few thousand of the ~20,000 human proteins are considered druggable [5]. Druggability assessments guide target selection and ranking in early discovery; however, many current AI evaluation benchmarks ignore druggability and treat every pocket as equally suitable for drug discovery.

Ignoring druggability in evaluation has serious consequences. A generative model may achieve a favorable docking score simply by generating large, hydrophobic molecules that fill any pocket [25]. If some of the pockets in a benchmark are intrinsically undruggable, high docking scores for those pockets are meaningless and may encourage the design of compounds that are unlikely to be viable

Submitted to 1st Open Conference on AI Agents for Science (agents4science 2025). Do not distribute.

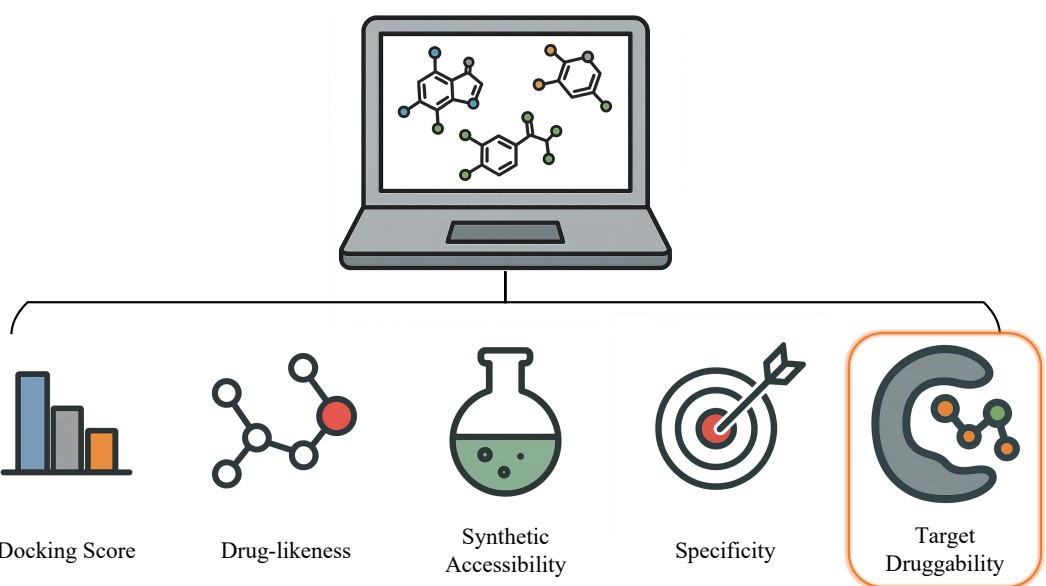

Figure 1: The evaluation of AI-driven structure-based drug design. We propose to incorporate **target druggability** into this evaluation framework.

drugs. Thus, there is a need to rethink evaluation protocols for structure-based generative models to account for the underlying druggability of targets. Recent perspectives also emphasize that the definition of druggability is itself evolving in the AI era. For instance, [3] highlights how machine learning can uncover new classes of druggable proteins by combining structural, sequence, and systems-level data. Classical definitions of druggability, based on geometric and physicochemical heuristics, are now being extended by AI approaches that leverage multi-modal biological knowledge, ranging from proteome-scale embeddings to network-based disease associations. This broadening of scope underscores why evaluation benchmarks that ignore druggability risk becoming detached from contemporary discovery practices.

In addition to these challenges, there is a growing awareness that modern SBDD must consider not only the ability to bind a target but also the broader pharmaceutical context. Molecules generated by AI models need to satisfy *druglikeness*, *synthetic accessibility*, and *specificity*. Experimental success stories—such as HIV protease inhibitors, kinase inhibitors and antibiotics identified through rational design—illustrate the potential of SBDD when the right targets are chosen [4]. Yet many clinical failures trace back to poor target selection or pockets that cannot be drugged.

To bridge the gap of aligning machine-learning evaluation with the realities of therapeutic discovery, we argue that generative models should not be judged solely on ligand-based metrics like docking scores in isolation; rather, these metrics need to reflect the underlying druggability of the target, as shown in Figure 1. To that end, we propose augmenting existing datasets like CrossDocked2020 [9] with continuous druggability scores and weighting generative performance accordingly. This refinement is expected to encourage models that prioritize genuinely tractable pockets and dissuade those that exploit bias in undruggable sites.

Looking ahead, embedding druggability into the AI-driven SBDD pipeline opens several opportunities. As more accurate pocket-scoring tools and experimental data become available, druggability-aware benchmarks could be expanded to cover a broader range of targets and conditions. The community could also explore models that jointly learn to predict druggability and generate ligands, ensuring that both target feasibility and molecular design evolve together. Ultimately, incorporating druggability should help steer generative algorithms toward compounds with a higher likelihood of clinical success, making this an important direction for future research and development.

## 2 Background

### 2.1 Structure-based Drug Design

Traditional SBDD workflows involve identifying a target protein, determining its 3D structure, locating a binding site and designing ligands iteratively. Computational techniques used in SBDD include structure-based virtual screening, molecular docking and molecular dynamics simulations [4]. Over the past decade, machine-learning models have been developed to predict binding affinities, model protein-ligand interactions and generate novel ligands [1, 15]. Deep learning models represent pockets as point clouds or graphs and learn features that capture spatial arrangements of chemical properties, including auto-regressive models [21, 13], diffusion models [10, 11] and flow models [22].

Evaluating these methods requires benchmarks that contain protein structures, binding pockets, and either known ligands or predictions derived from docking. CrossDocked2020 is currently the most widely used benchmark for pose prediction and generative design. It contains 13,839 unique ligands, 2,922 receptor pockets and ~22.6 million docked poses; about 41.9% of ligands have affinity data [9]. The dataset generates negative examples by cross-docking ligands into non-cognate pockets. For generative tasks of SBDD, benchmarks have been proposed to focus on docking scores (often from AutoDock Vina [23]): for example, the benchmark proposed by [8] uses the mean docking score for assessment. Besides, CBGBench [18] evaluates generative models within protein-ligand binding graphs, stressing relational reasoning. Tartarus [20] emphasizes realistic drug-design tasks, integrating pharmacokinetic constraints. Durian [19] provides a large-scale 3D molecular generation platform, enabling fair comparison across architectures. [27] recently questioned whether 3D methods consistently outperform 2D approaches, underscoring that methodological diversity remains essential. Collectively, these benchmarks demonstrate that while docking score remains the dominant evaluation criterion, there is growing interest in incorporating broader aspects of molecular feasibility—a trend our proposal seeks to extend by explicitly embedding druggability into evaluation.

### 2.2 Druggability and its Quantification

The concept of druggability emerged to prioritize proteins that can be modulated by small molecules. A druggable protein possesses a pocket whose shape and physicochemical properties complement drug-like molecules [5]. Several computational strategies exist to assess druggability:

- **Experience-based methods** rely on knowledge that members of the same family (e.g., GPCRs or kinases) have been successfully targeted by drugs. While useful, this approach may miss novel druggable proteins in uncharacterized families.

- **Ligand-based methods** infer druggability from known endogenous or synthetic ligands. The presence of a high-affinity ligand indicates that a suitable binding site exists, but this fails when no ligands are known.

- **Structure-based methods** analyze pocket geometry (size, depth, curvature), hydrophobicity and electrostatics. Geometry-based binding site predictors achieve ~74% success in identifying pockets [28]. Energy-based methods place probes around the pocket to estimate binding energies. Tools such as PockDrug [7], DrugPred [16], P2Rank [17] and DoGSiteScorer [24] combine these descriptors with machine-learning classifiers. These methods typically output continuous druggability scores or probabilities. Moreover, one-class learning approaches avoid explicitly defining the "non-druggable" class and learn the support of druggable pockets from positive examples [2].

- **Sequence-based methods** infer druggability from sequence motifs or protein-protein interaction networks. Machine-learning models using sequence features have been developed but often suffer from small training datasets and uncertain labels [12].

- **AI-driven methods.** In recent years, artificial intelligence has introduced a paradigm shift in druggability prediction. Unlike traditional structure- or sequence-based approaches, AI models integrate diverse feature sets, including 3D pocket descriptors, protein sequence embeddings, and systems biology context such as protein-protein interaction networks. For example, the DrugProtAI framework [12] applies robust ensemble learning and feature engineering to predict protein druggability with improved sensitivity and specificity, even

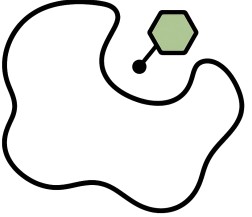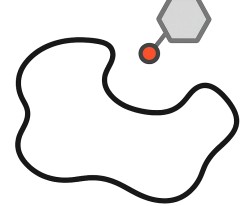

Druggable Targets

- Deep pocket
- Hydrophobic surface
- Well-defined cavity

*Examples:*
Kinases, GPCRs

Undruggable Targets

- Flat surface
- Polar site
- No defined cavity

*Examples:* KRAS, p53,
Myc, Phosphatases

Figure 2: Druggable vs. undruggable targets.

when structural data are limited. Graph neural networks and transformer-based sequence encoders have also been applied to infer cryptic or allosteric binding sites that might escape conventional predictors. While these methods improve coverage of the "dark proteome," they bring new challenges in terms of interpretability and potential bias from noisy training datasets. Together, they suggest that druggability should no longer be viewed as a static property but as a dynamic prediction informed by AI across multiple biological scales.

Assessment tools highlight that druggable pockets tend to be large, deep and hydrophobic [5]. However, the lack of consensus on non-druggable examples and the dynamic nature of pockets complicate predictions [2].

**Examples of druggable and undruggable targets.** Understanding concrete examples helps illustrate why druggability matters. **Druggable targets** often belong to protein families with well-defined pockets that accommodate small molecules. Protein kinases are the classic example of druggable targets: they possess an ATP-binding pocket that is deep and conserved, and dozens of kinase inhibitors have been approved for clinical use. Indeed, reviews emphasize that kinases are a representative class of druggable targets [26]. G-protein-coupled receptors (GPCRs) form another large family of druggable proteins; their seven-transmembrane architecture presents extracellular binding pockets that are highly amenable to modulation, and many marketed drugs act on GPCRs.

By contrast, **undruggable targets** lack obvious pockets or have surfaces that are flat, polar, or involved in protein-protein interactions. A typical example is KRAS, a small GTPase that was long considered undruggable because its surface lacks a defined pocket and its shallow binding site has undesired polarity [26]. Although a covalent inhibitor (sotorasib) has recently been approved for a specific KRAS mutation, the general class of RAS proteins remains difficult to drug. Phosphatases—enzymes that remove phosphate groups—are structurally similar within families, making it challenging to achieve specificity; low specificity and associated side effects hinder drug discovery [26]. Transcription factors such as p53 and Myc regulate gene expression and are involved in numerous diseases. Their structural heterogeneity and lack of tractable binding sites mean that conventional small molecules cannot easily bind them. Finally, protein-protein interaction (PPI) interfaces with flat surfaces, such as those in the B-cell lymphoma-2 (Bcl-2) family and intrinsically disordered proteins, are also considered undruggable [26]. These examples underscore the diversity of undruggable targets and highlight the need for evaluation protocols that penalize models for focusing on pockets that are unlikely to yield drug-like modulators.

Importantly, the boundary between druggable and undruggable targets is increasingly fluid as AI-based analyzes uncover new opportunities. For instance, cryptic binding pockets in KRAS and Myc—once paradigmatic undruggable proteins—have been identified using machine-learning-guided structural mining and molecular dynamics simulations [12, 26]. Similarly, AI-driven discovery of covalent inhibitors and allosteric modulators has begun to shift long-standing assumptions about RAS proteins and transcription factors. Moreover, degrader strategies such as PROTACs, aided by AI in

linker and degrader design, provide avenues to target proteins previously considered inaccessible. These developments illustrate that undruggability is not absolute but context-dependent, and reinforce the need for evaluation frameworks that can adapt to changing definitions of target tractability.

**Limitations of druggability metrics.** Despite advances, druggability assessment remains imperfect. First, the absence of reliable negative datasets makes it difficult to robustly define "undruggable" pockets; many are simply untested rather than truly intractable [2]. Second, static crystal structures cannot capture conformational dynamics, such as cryptic or transient binding sites, which AI and enhanced sampling methods are only beginning to uncover. Third, existing predictors may overweight hydrophobicity, leading to false positives for shallow hydrophobic cavities. Finally, AI-driven methods such as DrugProtAI [11] rely on training data from known druggable targets, which may bias predictions against novel protein classes. These limitations caution against treating druggability as a binary label and motivate our proposal for probabilistic, continuously valued scores.

# 3    Methodology: Incorporating Druggability into SBDD Evaluation

Existing SBDD benchmarks evaluate a model's ability to rank or generate ligands based on docking scores or pose accuracy. These metrics implicitly assume all pockets are equally viable drug targets. To incorporate **druggability** into evaluation, we propose the following evaluation protocol, based on CrossDocked2020 [9]:

**A. PDB-to-pocket mapping and preparation.** CrossDocked provides receptor coordinates and pocket definitions derived from the Pocketome. For each pocket, we extract the coordinates of residues within a certain radius around the ligand binding site. Before analysis, we remove crystallographic waters and keep counterions consistent with the original CrossDocked protocol.

**B. Druggability scoring.** We compute a continuous druggability score for each pocket using a state-of-the-art predictor (e.g., PockDrug [7] or DrugPred [16]). These tools accept the pocket coordinates and return a probability that the pocket is druggable. If multiple models are available, we could average their predictions to reduce variance. Because not all pockets are known to be druggable or undruggable, using a probabilistic score rather than a binary label allows a smooth weighting.

In addition to established predictors like PockDrug and DrugPred, advanced AI-based druggability predictors are expected to be incorporated into the scoring step. Such models may capture not only static pocket geometry but also dynamic and functional determinants of druggability, including protein family patterns and disease associations. A practical strategy would be to compute both traditional structure-based scores and AI-predicted probabilities, then combine them—either through weighted averaging or multi-criteria optimization—when calculating the druggability scores. This hybrid approach allows benchmarks to remain grounded in physical chemistry while also reflecting AI-driven redefinitions of what constitutes a druggable pocket. As AI models evolve, their predictions could be dynamically updated, ensuring that benchmarks capture the expanding frontiers of tractable target space.

**C. Reweighting of evaluation metrics.** Let $s_i$ denote the druggability score for pocket $i$, scaled to $[0, 1]$. In generative design tasks, models are typically evaluated only by ligand-based metrics such as the docking score. Based on this, we propose a **druggability-weighted docking score**:

$$\text{score}_{\text{weighted}} = \frac{\sum_i s_i \bar{D}_i}{\sum_i s_i}, \tag{1}$$

where $\bar{D}_i$ is the mean docking score of all generated ligands for pocket $i$. A higher $s_i$ gives more weight to pockets that are more druggable. This weighting emphasizes generation of good binders for realistic targets and reduces the influence of undruggable pockets. Metrics of molecular diversity can, however, be reported separately and remain unweighted. [6, 14].

**D. Thresholding and benchmark splits.** To facilitate comparison with current benchmarks, we create subsets of CrossDocked2020 at different druggability thresholds (e.g., 0.2, 0.5, 0.8). The high-druggability subset includes pockets with $s_i > 0.5$ and represents realistic targets; the low-druggability subset can serve as negative controls or test a model's ability to avoid undruggable sites.

**E. Calibration and validation.**  Because druggability predictors may themselves be biased, we recommend validating the reweighted benchmark using known drug-target pairs: evaluate whether pockets with high $s_i$ correspond to proteins with approved drugs and adjust scoring functions accordingly. Further, one should test whether models that perform well under weighted metrics also yield compounds with favorable drug-likeness and high synthetic accessibility.

**F. Analysis of model performance after weighting.**  After integrating druggability scores into the evaluation, researchers should analyze how different model classes perform across the druggability spectrum. For example, diffusion models conditioned on pocket geometry [11] may excel at generating ligands for highly druggable pockets because the latent space can capture well-defined cavities. Conversely, graph-based retrieval-augmented models or 1D/2D genetic algorithms might generate structurally diverse molecules that occasionally fill low-druggability or atypical pockets, leading to higher scores in the unweighted setting but being penalized under our weighting. Models that perform well at undruggable sites might be exploiting spurious correlations (e.g., generating large hydrophobic molecules), which could translate into poor specificity or toxicity. A comparative analysis can thus reveal the superiority of some methods at realistic targets (high druggability) and highlight the risks of overfitting to undruggable cavities. Such insights will guide future model development and help prioritise architectures that generalise across druggable targets while avoiding pathological behaviours.

## 4   Conclusion and Discussion

In summary, our work emphasizes that druggability is a critical variable missing from current SBDD evaluation protocols. By integrating druggability into CrossDocked2020, we aim to provide a more realistic assessment of generative models. Weighted metrics focus attention on pockets where medicinal chemistry is most likely to succeed and discourage the generation of large hydrophobic ligands that fill any cavity. Our approach also introduces heterogeneity into the benchmark, allowing researchers to compare model performance across different druggability regimes.

Several limitations should be acknowledged. Druggability predictors themselves rely on training datasets of known druggable pockets and may misclassify novel types of pockets; energy-based predictors may overestimate pockets that favor hydrophobic fragments. Moreover, weighting metrics by druggability may reduce the influence of novel but low-probability targets, potentially discouraging exploration of innovative chemistries. It is important to maintain separate analyzes for high- and low-druggability subsets rather than exclude the latter entirely.

Future work should explore joint learning of druggability and ligand design. Multitask models could simultaneously predict pocket druggability and generate ligands, allowing the model to allocate resources appropriately. Improved datasets with experimentally validated undruggable pockets would reduce reliance on proxies. Finally, evaluation should incorporate additional factors such as specificity, ADMET properties and synthetic accessibility. Nonetheless, integrating druggability into benchmark evaluation represents a practical step toward aligning AI-driven SBDD with real-world drug discovery.

Looking ahead, embedding AI-driven redefinitions of druggability into structure-based drug design promises to further align generative evaluation with therapeutic reality. Classical metrics based solely on pocket geometry risk excluding cryptic or context-dependent sites that AI now reveals as druggable. Conversely, AI methods risk overestimating tractability if benchmark design fails to enforce chemical realism. Thus, future SBDD benchmarks should adopt a hybrid paradigm: structural druggability scores for stability and interpretability, coupled with AI-derived predictions for sensitivity and discovery of novel opportunities. Multitask frameworks that co-train generative models to optimize both ligand fit and AI-predicted target feasibility represent an especially promising direction. Ultimately, this synthesis of structure-based heuristics with AI-derived insights could redefine not just how we evaluate generative models, but also how we conceptualize the druggable genome itself.

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
